

# A 'post-honeymoon' measles epidemic in Burundi: mathematical model-based analysis and implications for vaccination timing

Katelyn C. Corey[1] and Andrew Noymer[2]

[1] Fielding School of Public Health, University of California, Los Angeles, CA, United States
[2] Department of Population Health and Disease Prevention, University of California, Irvine, CA, United States

## ABSTRACT

Using a mathematical model with realistic demography, we analyze a large outbreak of measles in Muyinga sector in rural Burundi in 1988–1989. We generate simulated epidemic curves and age × time epidemic surfaces, which we qualitatively and quantitatively compare with the data. Our findings suggest that supplementary immunization activities (SIAs) should be used in places where routine vaccination cannot keep up with the increasing numbers of susceptible individuals resulting from population growth or from logistical problems such as cold chain maintenance. We use the model to characterize the relationship between SIA frequency and SIA age range necessary to suppress measles outbreaks. If SIAs are less frequent, they must expand their target age range.

# INTRODUCTION

Measles is a viral disease of worldwide public health importance despite enormous reduction in incidence and mortality since the 1980s (*Otten et al., 2003*; *Otten et al., 2005*; *Brenzel et al., 2006*; *Perry et al., 2014*). Foremost among the problems of measles control is the *post-honeymoon epidemic*, occurring when susceptibles accumulate in a population despite relatively good vaccine coverage (*Cutts & Markowitz, 1994*). These outbreaks are not limited to developing countries; *Pyle (1973)* documents a post-honeymoon outbreak in the USA, seven years after the introduction of vaccination. Such epidemics were especially a problem in the late 1980s and early 1990s (*Gindler et al., 1992*; *Mulholland, 1995*), but continue to this day, particularly in the presence of "antivax" sentiment (*Majumder et al., 2015*), and health system interruptions (*Takahashi et al., 2015*). Mathematical models can play a role in understanding epidemics, particularly when natural equilibria are perturbed by vaccination. The outbreak we model is the 1988–89 post-honeymoon epidemic in Muyinga sector, Burundi (*Chen et al., 1994*). One goal is to make outbreak-avoiding recommendations for measles vaccine policy in high growth rate populations.

Before the introduction of vaccination in 1963 and thereafter (*Katz & Gellin, 1994*), measles was ubiquitous; virtually everyone acquired measles, usually in childhood.

Corresponding author
Andrew Noymer, noymer@uci.edu

Therefore, the annual number of infections in pre-vaccination populations was approximately equal to the size of the birth cohort, discounted by population growth, since measles occurs on average a few years after birth, and also by the mortality of those who do not live long enough to become infected. Immunity following natural measles infection is both high and life-long, so serum antibody is a reliable marker of current or past measles infection. Measles is highly contagious: *Hope Simpson (1952)*, studying household contacts in Cirencester, England, derived a susceptible-exposure attack rate of 75.6%.

Measles has been a favorite topic for mathematical modelers (*Fine & Clarkson, 1982a*; *Fine & Clarkson, 1982b*; *Fine & Clarkson, 1983*; *Bjørnstad, Finkenstädt & Grenfell, 2002*). Its airborne transmission route does not require detailed specification of different types of contact between individuals. Moreover, peak infectiousness occurs during the prodrome period, before the outbreak of the rash (*Hamborsky, Kroger & Wolfe, 2015*), which means that the assumption of mixing between susceptibles and infecteds is reasonable. There are thought to be no subclinical cases. Given that measles is a potential eradication target (*Cutts & Steinglass, 1998*; *Strebel et al., 2011*; *Christie & Gay, 2011*; *Goodson et al., 2012*; *Sniadack & Orenstein, 2013*), understanding its epidemiology in a variety of demographic settings is desirable.

The results of models like ours have influenced vaccination policy, both in terms of vaccination age and in scheduling supplementary immunization activities (SIAs) (*Bart et al., 1983*; *Ramsay et al., 1994*; *Gay & Miller, 1995*; *Gay et al., 1997*). We analyze data on a post-honeymoon measles outbreak, using a partial differential equation (PDE) epidemiologic model with explicit demography. Our results underscore the need for high vaccine coverage and the use of SIAs to rectify shortfalls in routine vaccination. Based on the model, we quantify trade-offs between frequency and age range of SIAs necessary to suppress measles outbreaks. We also demonstrate a counterintuitive result that higher-growth populations have slightly lower vaccination requirements, although we note that this is not of policy importance.

## MATERIALS AND METHODS

*Schenzle (1984)* was the first application of *Hoppensteadt*'s (*1974*) age-dependent epidemic model to measles. *McLean (1986)*, *McLean & Anderson (1988a)* and *McLean & Anderson (1988b)* furthered the development, as did *John (1990a)*, *John (1990b)* and *Tuljapurkar & John (1991)*. We apply a model like those, to surveillance data from the 1988–89 Muyinga sector (Burundi) measles outbreak. This epidemic was described by *Chen et al. (1994)*, and was featured in a public health training manual (*Chen & Morinière, 1993*).

We implement an age-structured MSEIR model, which is like the standard SEIR model with the addition of a maternal class (newborns protected by maternal antibodies) (*McLean & Anderson, 1988a*; *McLean & Anderson, 1988b*). The classes are: maternal, susceptible, latent, contagious, and immune; abbreviated herein as $M$, $S$, $L$, $C$, and $Z$. We use realistic demography tailored to rural Burundi, and the model assumes a demographically stable population (i.e., constant population growth rate with unchanging age structure, *Coale, 1972*). Figure 1 is a model schematic, and the model is specified in Eqs. (1)–(5) (p. 4),

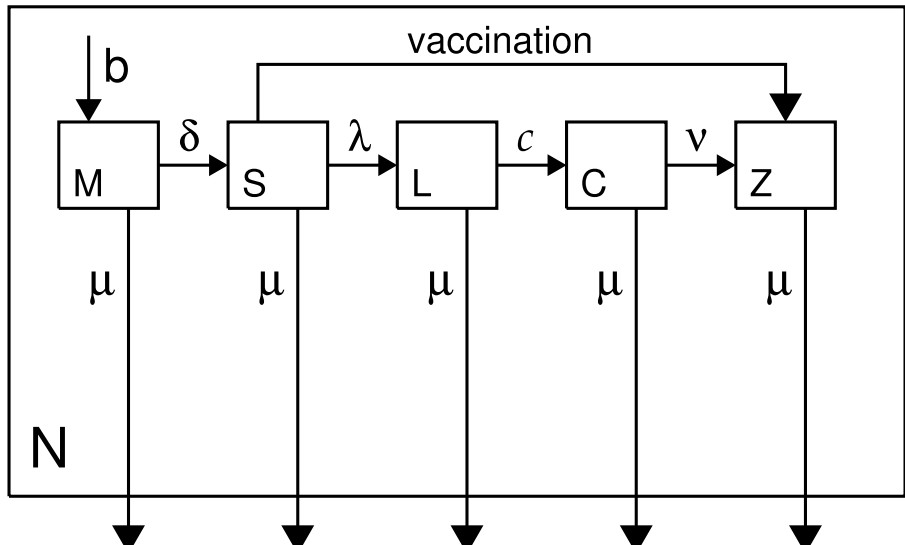

**Figure 1  Model schematic.** Classes are as described in the text and in Eqs. (1)–(5) (p. 4); *N* denotes total population.

which are solved numerically (*Eriksson et al., 1996*) with the IDL language, v.8.4 (Exelis Inc., Boulder CO), using Euler's method with an age/time step of 2.5 days. The type of nonlinear PDE in age and time specified on p. 4 is similar to many mathematical models in epidemiology (cf. *Anderson & May, 1991*; *Brauer & Castillo-Chavez, 2012*).

We use a mean duration of the latent period of 10 days, and mean length of the contagious period of 7 days. The variances of the duration of the latent and infectious periods are low (*Conlan et al., 2010*). No measles-specific mortality is included in the model. However, measles fatality does not greatly affect transmission dynamics because most deaths occur coincidentally with, or following, the desquamation of the rash, which marks the end of the contagious period (*Clements et al., 1993*).

Maternal represents the class who are immune due to the persistence of trans-placentally acquired antibodies (*Cáceres, Strebel & Sutter, 2000*). A six-month protected period is assumed, a slight oversimplification (*Williams, Cutts & Dye, 1995*). Attenuation of maternal antibody is closely related to the so-called window problem (*Dabis et al., 1989*; *Cutts & Markowitz, 1994*; *Sakatoku et al., 1994*; *Hartter et al., 2000*). Breastfeeding does not confer direct (immunological) protection against measles (*Adu & Adeniji, 1995*; *Oyedele et al., 2005*). Susceptibles are the population, age ≥6 months, transferring to the latent class at rate λ, or until vaccination-induced immunity provides a move to the immune class. Vaccination is shown in the schematic but it is not part of Eqs. (1)–(5) because it is exogenous to the epidemiology. Successful vaccination in this model is assumed to provide life-long immunity (no boosting).

The transition rate from susceptible to latent is the force of infection, $\lambda = \beta(C/N)$, where $\beta$ is a constant (Eqs. (6)–(8)). Beta combines a social process with a biological one: the mixing of the population with itself; and the probability that susceptible-infected contact will result in a new infection. If we assume, as *Wilson & Worcester (1941)* did in their early

modeling of the force of infection for measles, that contact between an infected and a susceptible always results in infection, then $\beta$ is simply a constant regulating population mixing.

**Model equations:**

$$\frac{\partial M}{\partial a} + \frac{\partial M}{\partial t} = -(\delta(a-\zeta)+\mu(a))M(a,t) \tag{1}$$

$$\frac{\partial S}{\partial a} + \frac{\partial S}{\partial t} = \delta(a-\zeta)M(a,t)-(\lambda(t)+\mu(a))S(a,t) \tag{2}$$

$$\frac{\partial L}{\partial a} + \frac{\partial L}{\partial t} = \lambda(t)S(a,t)-(c+\mu(a))L(a,t) \tag{3}$$

$$\frac{\partial C}{\partial a} + \frac{\partial C}{\partial t} = cL(a,t)-(v+\mu(a))C(a,t) \tag{4}$$

$$\frac{\partial Z}{\partial a} + \frac{\partial Z}{\partial t} = vC(a,t)-\mu(a)Z(a,t) \tag{5}$$

Boundary conditions:

$$M(0,t)=b(t)$$

$$\mu(\omega)=\infty$$

**Notation in Eqs. (1)–(5):**

| Symbol | Quantity |
|---|---|
| $M$ | Class protected by maternal antibody |
| $S$ | Susceptible class |
| $L$ | Latent class |
| $C$ | Contagious class |
| $Z$ | Permanently immune class |
| $a,t$ | age, time |
| $b(t)$ | births. $\partial b/\partial t = r \cdot b(t)$. $r$ is the population growth rate. |
| $\delta(\cdot)$ | Dirac function |
| $\zeta$ | age at which the protection of maternal antibodies ends |
| $\mu(a)$ | force of mortality |
| $\beta$ | mass-action constant (see Eqs. (6)–(8)) |
| $\lambda(t)$ | force of infection $\lambda(t)=\beta \cdot C(t)/N(t)$ |
| $c$ | rate at which latents become contagious, 0.1 days$^{-1}$ |
| $v$ | recovery rate, 0.143 days$^{-1}$ |

**Lambda, the force of infection:**

$$\lambda(t)=\beta\frac{C(t)}{N(t)}=\beta\int_0^\omega C(a,t)da \left/ \int_0^\omega N(a,t)da \right. \tag{6}$$

$N$ is the sum of all epidemiological classes $(M,S,L,C,Z)$; $\omega$ is the oldest age. In the model, $\beta$ is fixed, while $\lambda(t)$ varies; $\beta$ is derived from equilibrium conditions $(\partial\lambda/\partial t \equiv 0)$, as follows. A version of the model is run in which:

$$\lambda(t)\equiv\lambda^*=(\bar{a}-\zeta)^{-1} \tag{7}$$

where $\bar{a}$ is the mean age of infection (pre-vaccination; exogenous of the model), and $\zeta$ is the age at which protection from maternal antibodies ends. Then:

$$\beta = \lambda^* \frac{N(t)}{C(t)} = \lambda^* \int_0^\omega N(a,t)da \bigg/ \int_0^\omega C(a,t)da \qquad (8)$$

where Eq. (8) is calculated once $N(t)/C(t)$ reaches a stable equilibrium (see also main text).

Because $\beta$ is constant, the force of infection, $\lambda$, changes when the ratio $C(t)/N(t)$ changes. This implies that during an epidemic there are no behavior changes that affect how infecteds and susceptibles mix. The logic of Eqs. (6)–(8) is as follows. Before vaccination, measles was endemic; vaccination introduces epidemic cycles by perturbing the equilibrium. Thus, vaccination can create epidemics, although in the long run the total disease burden of measles declines. In the pre-vaccine era, endemic equilibrium holds, so the force of transmission, $\lambda$, is constant. Call this endemic force of transmission $\lambda^*$. This implies an exponential distribution, with the average age of measles infection given by $1/\lambda^*$, or equivalently $\lambda^* = 1/\bar{a}$, where $\bar{a}$ is the mean age of infection. We adjust for the period of maternal antibody protection from birth to age $\zeta = 6$ months.

Much in the model depends upon $\beta$, which is estimated as follows. Serological data collected during the pre-vaccination (endemic) era permit estimation of $\bar{a}$, and therefore $\lambda^*$, the endemic force of infection. We then run the model, with $\lambda \equiv \lambda^*$. This leads to endemic dynamics, with an equilibrium proportion infected, $(C/N)^*$. Recall that $\lambda = \beta(C/N)$. Since we can get $(C/N)^*$ from the equilibrium simulation, and we know $\lambda = \lambda^* = 1/(\bar{a} - \zeta)$, we can solve for $\beta$. Without the complication of realistic demography, $\beta \approx R_0(\nu + \mu)$, where $R_0$ is the net reproductive rate of measles (*May & Anderson, 1985*) and $\mu$ is the force of mortality (non-age-dependent, hence without realistic demography). However, the simulated equilibrium process, as described, takes the demography into account, and does not require external estimates of $R_0$ (which can vary from population to population).

The above depends on getting an estimate of $\bar{a}$. We have not found any estimates of the mean age of infection for Burundi in the literature, but Table 1 reviews similar figures for other countries in Africa. Pre-vaccination estimates must be used here because vaccination interferes with the natural epidemiology of measles. Using pre-vaccination data is a good way to get a reliable measure of population mixing under the assumption of constant $\lambda$ (*De Jong, Diekmann & Heesterbeek, 1995*). An estimate of $\bar{a} = 30$ months was chosen based on the available data. Since 30 months less six months of antibody protection is 24 months, we have $\lambda^* = 0.5 \text{ yr}^{-1}$. Serology naturally reports percentiles of the cumulative distribution function of measles exposure by age, and, thus, medians not means (cf. Table 1); assuming exponential distributions (which is reasonable in the pre-vaccination era), the conversion is mean = median/log(2). The model does not use age-dependent transmission rates (*Anderson & May, 1991*; *Eichner, Zehnder & Dietz, 1996*). To the best of our knowledge, no data exist to estimate such rates for Burundi. In any case, the age-independent $\beta$ performs well relative to the data (cf. below), indicating that equal mixing of all age groups is a reasonable assumption for rural Burundi.

**Table 1** Published estimates of median age of measles infection in Africa.

| Population, date | Age (months) | Method[a] | Population-level Vaccination |
|---|---|---|---|
| Casablanca, non-European, 1953 | 24 | SS | none |
| Dakar, 1957 | ≈12 | SS | none |
| Rural Sénégal, 1957 | 12–24 | SS | none |
| Ilesha, Nigeria, 1962 | <17 | HO | none |
| Morocco ("average age"), 1962 | 24–36 | n/i | none |
| Ilesha, Nigeria, 1963–64 | ≈20 | HO | none |
| Sénégal ("average age"), 1964 | 12–24 | n/i | none |
| Ghana ("average age"), 1960–68 | 24–36 | n/i | none |
| Lagos, 1970 | 15 | CR | none |
| W. & Cent. Africa, dense urban, 1971 | 14 | CR | none |
| W. & Cent. Africa, urban, 1971 | 17 | CR | none |
| W. & Cent. Africa, dense rural, 1971 | 22 | CR | none |
| W. & Cent. Africa, rural, 1971 | 29 | CR | none |
| W. & Cent. Africa, isolated rural, 1971 | 48 | CR | none |
| Yaoundé, 1971 | ≈15 | CR | none |
| Yaoundé, 1975 | ≈20 | CRE | limited |
| Yaoundé, 1975 | 12–23 | CR | limited |
| Yaoundé, 1976 | 12–17 | SS | limited |
| Machakos, Kenya, 1974–76 | 30 | CRSS | low (≈25%) |
| Machakos, Kenya, 1974–77 | ≈31 | CRSS | low (≈25%) |
| Machakos, Kenya, 1974–81 | 42 | CR | increasing level |
| Moshi, Tanzania, no exact date | 24–36 | SS | none |
| Kinshasa, 1983 | 12–24 | CS | ≈60% coverage |
| Pointe-Noire, Congo, 1983 | 18 | HC | partially vaccinated |
| Pointe-Noire, Congo, 1985 | 20 | HC | 54% coverage |
| West Africa, n/s | 18 | n/i | no information |
| Rural Guinea-Bissau, n/s | 42 | CS | none |
| Rural Gambia, n/s | 60 | CS | none |
| Rural Sénégal, n/s | 42–60 | CS | none |
| Rural Somalia, n/s | 42 | CS | no information |
| Urban Guinea-Bissau, n/s | 24–30 | CS | none |
| Urban Zambia, n/s | 24–30 | CS | yes |
| Urban Sénégal, n/s | ≤24 | SS | before vax. programs |

**Notes.**

[a]**Key:** SS, serological survey/study; HO, hospital outpatients; n/i, no information; CR, case reports; CRE, case reports (epidemic); CRSS, case reports (some serology); CS, Community study/survey; HC, hospitalized cases.

**Sources:** *Anderson & May, 1985*; *Black, 1962*; *Boué, 1964*; *Cutts, 1990*; *Dabis et al., 1988*; *Foster, McFarland & John, 1993*; *Guyer, 1976*; *Guyer & McBean, 1981*; *Remme, Mandara & Leeuwenburg, 1984*; *Leeuwenburg et al., 1984*; *McBean et al., 1976*; *Morley, 1962*; *Morley, 1985*; *Muller et al., 1977*; *Taylor et al., 1988*; *Voorhoeve et al., 1977*; *Walsh, 1986*.

In the model with vaccination, the perturbed state is run for two years, and then vaccination begins, with a mass vaccination campaign as its opening salvo, as was the case in Muyinga in 1981 (*Chen et al., 1994*). The model is not designed to estimate the net reproductive rate ($R_0$). Nonetheless, our results show that the force of infection in the model (and, by implication, our estimate of $\lambda^*$) are in-line with conventional $R_0$

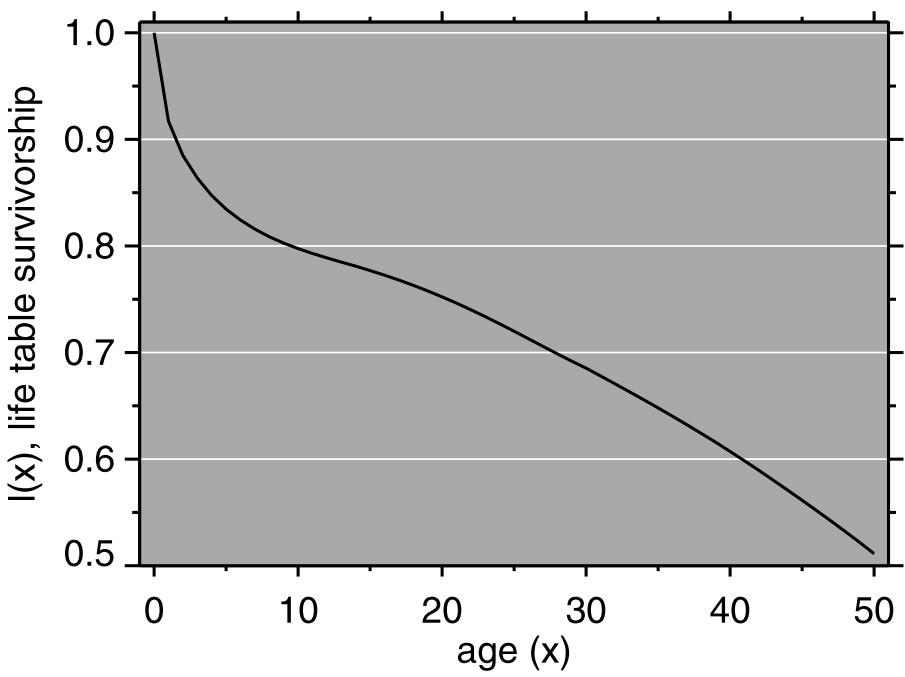

**Figure 2** Mortality model: life table survivorship curve.

estimates for measles virus. Using the first-order approximation $p^* = 1 - R_0^{-1}$, where $p^*$ is the herd immunity threshold (*Edmunds et al., 2000*), and extrapolating from our results, gives the interval estimate $6.7 < R_0 \leq 20$. This is wide, but our goal is to inform vaccine policy; estimating $R_0$ is outside our primary scope (see also *Heesterbeek, 2002* on $R_0$ and its strengths and limitations).

We used population data from the 1987 Burundi Demographic and Health Survey (DHS) (*Segamba et al., 1988*). Population growth in the model was 2.6% per year (*ibid.*). Using a Brass logit relational model life table (*Brass & Coale, 1968*), we estimated age-specific mortality rates from DHS mortality data. Mortality in Burundi, according to our fitting approach, resembles that of North level 14 for females (life expectancy at birth, $e(0) = 52.5$) and North level 15 for males ($e(0) = 51.4$) (*Coale & Demeney, 1983*). The male and female life tables were combined using the sex ratio at birth in Burundi (*Garenne, 2002*). The life table survivorship function is shown in Fig. 2. The starting total population was scaled according to the size of Muyinga district, which *Chen et al. (1994)* give as approximately 330,000 for 1988.

The proportion of susceptibles transferred to immune is the product of the vaccine coverage and the vaccine efficacy. There are two types of vaccination in the model: routine vaccination, and SIAs. In the case of routine vaccination, in the model it occurs upon reaching a certain exact age (9 months has been used throughout). This is a simplification; in the real world, infants 9–11 months are targeted. Routine vaccination is implemented at each time step. Supplementary immunization activities, on the other hand, vaccinate all children in a certain age band (subject to coverage limitations), but only once, at a single time step. In this model, the supplementary immunization activity (SIA) coverage was 70%

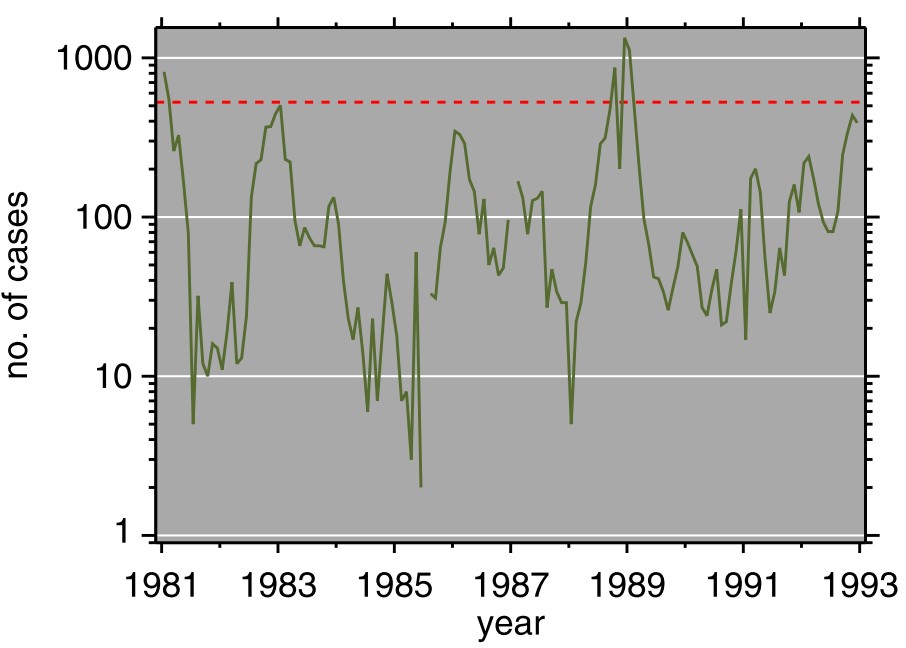

**Figure 3** **Measles incidence, Muyinga sector, Burundi, 1981–1992.** Red dashed line is epidemic threshold (mean + 1.96SD, after *Cullen et al., 1984*).

of the population between 9 and 23 months, with an assumed vaccine efficacy of 80%; this is based on the description of *Chen et al. (1994)* p. 187.

## RESULTS

Monthly routine measles surveillance data from the Expanded Programme on Immunization (EPI) in Burundi, 1981–92, are plotted in Fig. 3. Figure 4 is a scatterplot of measles and chickenpox in Muyinga sector. In tropical settings such as Burundi, without winter-summer cyclicality, diseases like measles and chickenpox are not typically in synchrony. Thus, unless induced by reporting effects, we do not expect co-movement between these unrelated diseases. In addition to the lack of relationship in Fig. 4, there is no significance ($p = .19$, two sided) in a Goodman–Grunfeld (*1961*) time series test for co-movement. The chickenpox data show that changes in measles incidence are not reporting artifacts.

In Fig. 3, there are some small outbreaks after the introduction of vaccination, but before the large post-honeymoon epidemic. Muyinga sector is small enough that long-term transmission of measles requires re-introduction from neighboring sectors (*Black, 1966*). However, if viral introductions occur when conditions are not ripe for a large epidemic, they only cause smaller outbreaks and perpetuate low-level transmission.

Figure 5 depicts the results of the model with vaccination, as an age × time × prevalence surface. The age distribution of measles is pushed upward by vaccination. The surface also has local maxima at relatively older ages, indicating that everyone who is not successfully immunized will contract measles sooner or later. Older children who missed immunization should not be forgotten by vaccination efforts; SIAs can fill the gap. Figure 6 gives the mean

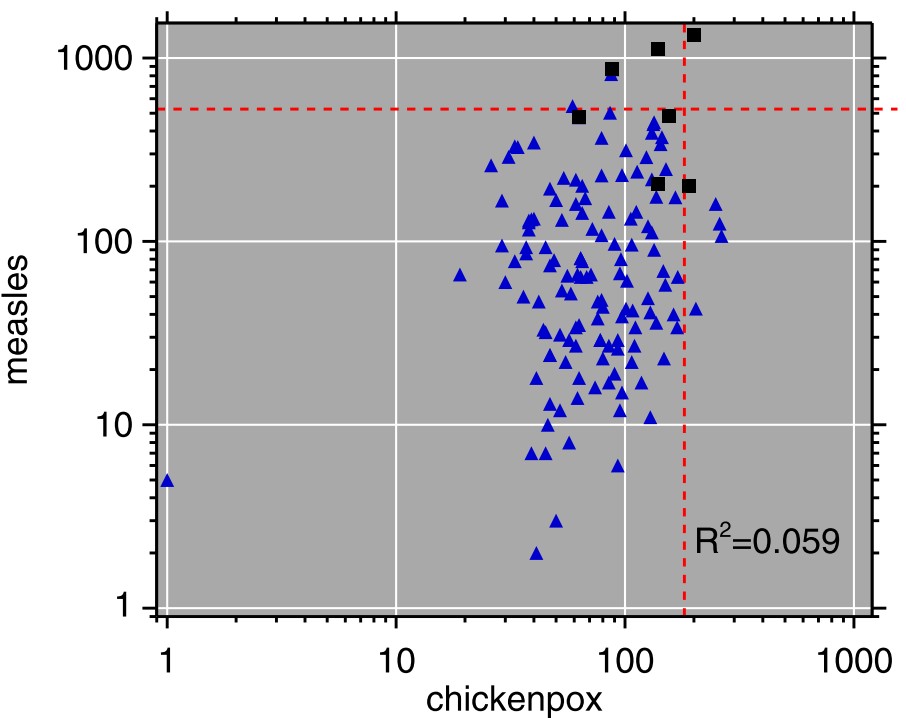

**Figure 4** **Log–log scatterplot of measles and chickenpox monthly incidence.** The 7 months centered on the post-honeymoon measles outbreak are plotted as black squares. Dashed lines indicate epidemic thresholds (see Fig. 3 caption). The month with 1 reported chickenpox case is likely a reporting error.

and standard deviation of the age of measles cases. Like the surface, this shows how both of these quantities move upward, also reinforcing the idea that as vaccination programs mature, attention should be paid to expanding their coverage, age-wise. The increase in age is relevant for mortality, because older children have lower case fatality rates (*Walsh, 1986*; *Cutts, 1990*; *Wolfson et al., 2009*). This assumes that post-honeymoon measles epidemics do not have higher case fatality rates, which has been suggested (*Garenne, Glasser & Levins, 1994*).

Table 2 presents the age distribution for the first post-honeymoon epidemic. Age-stratified data from Muyinga are only available during the outbreak, not as a longer time series. The simulated data are broadly consistent with the empirical data. The model reports no cases below 6 months because it cannot: this is the duration of maternal antibody protection. The modest number of cases below 6 months in the observed data suggests that the model assumptions regarding maternal antibody are reasonable. In general, the model age structure is a good qualitative match to that reported by *Chen et al. (1994)*.

Regarding vaccine policy, the role of models such as these is not only to simulate a specific outbreak but to allow counterfactual investigation. Figure 7 shows simulated time series for Muyinga, where net immunization (i.e., vaccine coverage times vaccine efficacy) is allowed to vary. This shows clearly that as coverage improves, the time until the first post-honeymoon outbreak gets longer and longer. Even when net immunization is as low as 75%, in a population the size of Muyinga, the incidence becomes fractions of a

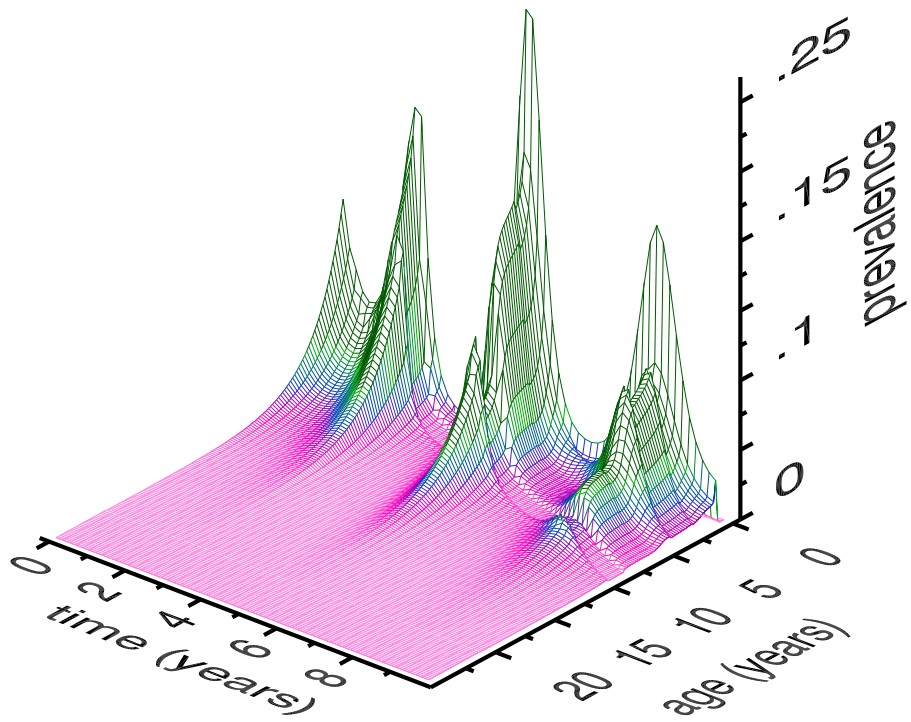

**Figure 5   Age × time × prevalence surface (class C of the model).** Model conditions are for a simulation of Muyinga sector, Burundi. Time zero represents two years prior to the introduction of vaccination. Thus, the peak on the far left is the end of pre-vaccine epidemics, and the first main trough is the honeymoon period. The central, largest, peak is the post-honeymoon epidemic. The vertical axis is to be interpreted as the height of the wireframe above an individual 1 × 1 square of age × time (in months). Summing over all ages, the total monthly prevalence represented by this surface ranges from 15.4 to 523.5 cases.

**Table 2   Age distribution of measles cases in post-honeymoon epidemic, model versus observed.**

| (months) Age | Model | (from *Chen et al., 1994*, p. 189) Observed |
|---|---|---|
| 0–5 | 0% | 5% |
| 6–11 | 23% | 27% |
| 12–23 | 32% | 24% |
| 24–35 | 27% | 19% |
| 36–59 | 18% | 25% |

person. Arguably, this could be seen as temporary elimination. However, it is clear, with net immunization as high as 85%, the population is still susceptible to a large outbreak upon reintroduction of the virus (e.g., from a neighboring province). In the absence of reintroduction, even the fractional cases eventually are enough to spark a post-honeymoon outbreak. The model does not incorporate demographic stochasticity, which imposes integer constraints (*Mollison, 1981*; *Snyder, 2003*). When net immunization is 95%, Fig. 7 shows that measles does not endogenously reappear on a 25-year horizon.

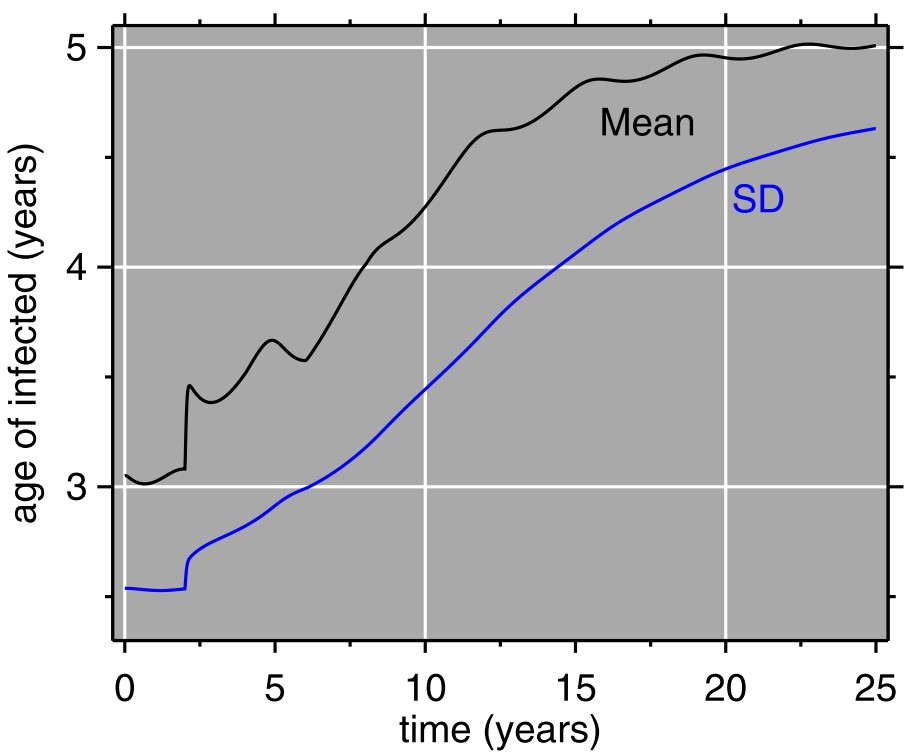

**Figure 6** **Model results: mean age and standard deviation (SD) of age of measles cases.** The horizontal axis (time) matches that of Fig. 5: vaccination is introduced at 2 years, accompanied by abrupt increases in the mean and SD of the age of measles cases.

Figure 8 is a heatmap showing the waiting time to occurrence of the first post-honeymoon epidemic, varying the net immunization rate and the population growth rate. The diagonal white band represents a 15-year waiting period, and the red region shows that whenever the net immunization rate is below 80%, post-honeymoon epidemics will occur in 15 years or less, regardless of the population growth rate. As noted, these arise from persistence of fractional numbers of cases in the inter-epidemic period, but also show that the population would be vulnerable in the event of measles virus introduction. The heatmap transitions abruptly to dark blue, indicating no endogenous reoccurrence of measles within 25 years.

Above 90% net immunization is required to achieve permanent suppression of post-honeymoon outbreaks. Without SIAs, this requires about 95% coverage with a 95% efficacious vaccine, which is a major challenge in rural areas of low-income countries where cold chain maintenance is difficult. When the population growth rate is higher, the vaccine coverage requirements appear slightly more lenient. This may seem counterintuitive, given that population growth drives the creation of new susceptible children. The bottom-heavy nature of population pyramids in high-growth societies drives the effect; everyone in the model age 6 months or less is immune. The higher the growth rate, the greater proportion of the population is immune through maternal antibodies. However, this has little practical importance, since the lenient tilt of the white band in Fig. 8 is more than offset by the challenges of vaccinating more and more children (in absolute numbers) after 6 months

 

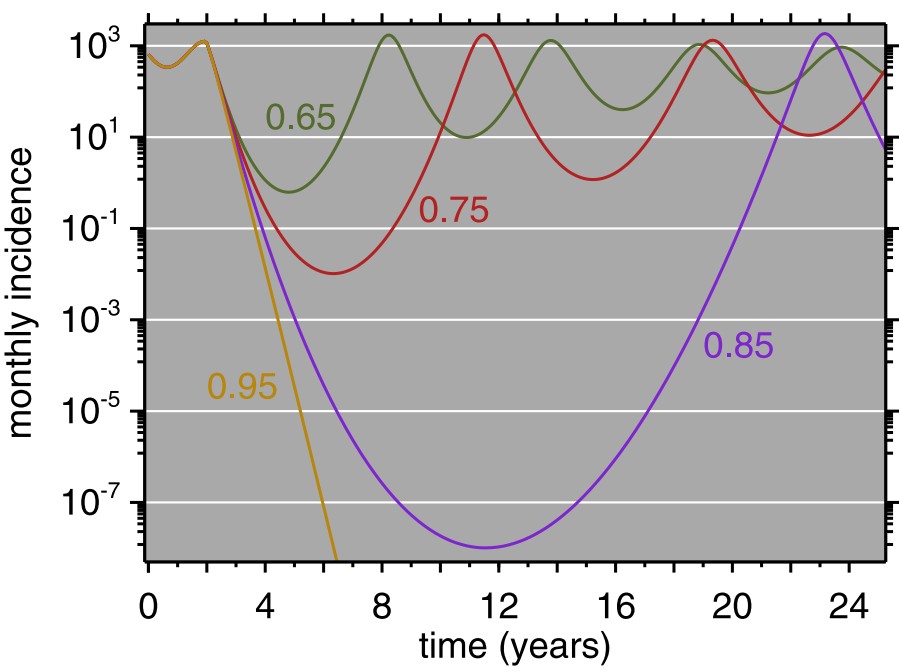

**Figure 7** **Model Results (all ages) for four levels of immunization, 0.65, 0.75, 0.85, 0.95.** Immunization equals vaccine coverage multiplied by vaccine efficacy. Vaccination is introduced after two years of simulation, at which point the four curves diverge.

of age, in growing populations. Indeed, in the real world, populations with 2% and higher growth rate have a large rural share, which presents its own obstacles to universal vaccination.

Supplementary immunization activities can work synergistically with routine vaccination (*Helleringer, Asuming & Abdelwahab, 2016*), especially when net immunization from routine vaccination is too low to provide good measles control. Low net immunization can be the product of shortfalls in coverage, or low vaccine efficacy as a result of cold chain problems. Even in simulated conditions of 65% routine vaccine coverage with a 75% effective vaccine (with the latter value chosen to reflect cold chain difficulties), good control can be achieved if routine vaccination is augmented by a suitable regime of SIAs. Our simulations assume SIA coverage is 70%, with 80% vaccine efficacy; we assume that SIAs can do a little better on efficacy than routine vaccination. For example, seroconversion rates are higher in older children, and SIAs cover older children on average, compared to routine vaccination. In our simulations, good control can be achieved with SIAs, but there is a trade-off between the frequency and the age range of SIAs. The longer time between SIAs, the wider the target age range must be to suppress outbreaks. The frequency-maximum age relationship is nonlinear, and is depicted graphically in Fig. 9 (p. 14). The dark blue region represents effective control, and it shows clearly that as the interval between SIAs increases, the upper bound of the target age range must increase (the lower bound is fixed at 9 months). *Verguet et al. (2015)* analyze similar situations. Very frequent SIAs are for all intents and purposes a different type of routine vaccination, thus we did not examine frequency greater than once every 24 months.

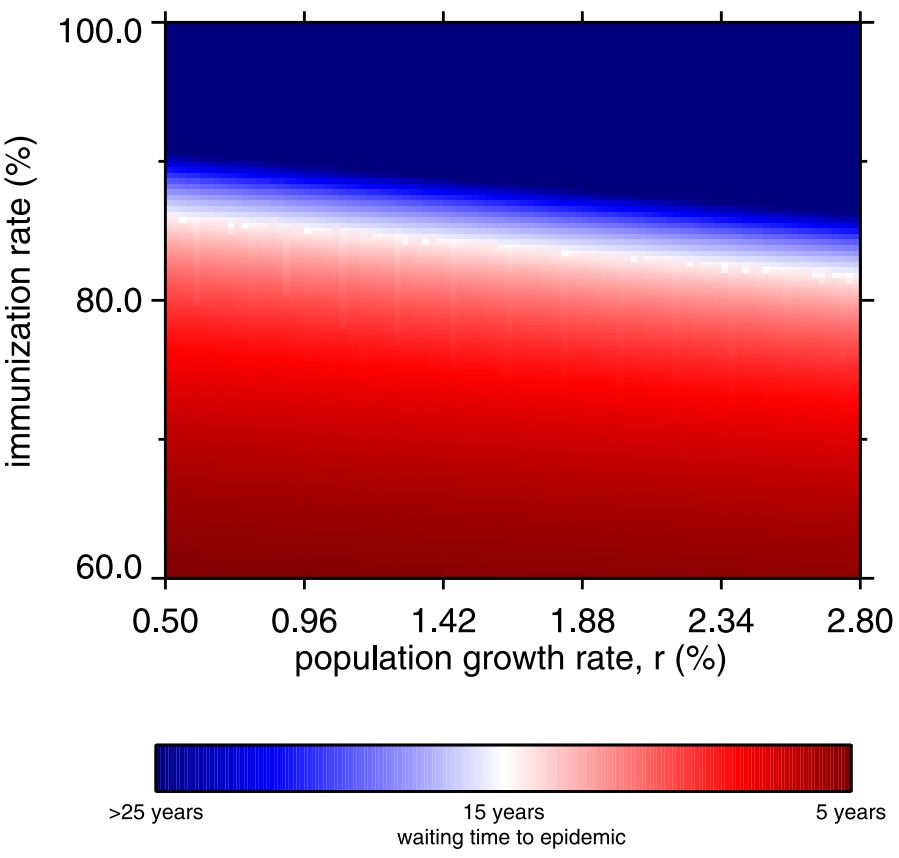

**Figure 8** **Heatmap, waiting time to occurrence of the first post-honeymoon epidemic, by net immunization rate and the population growth rate.** From 10,000 model runs.

## DISCUSSION & CONCLUSIONS

Since the 1980s, enormous strides have been made in measles control (e.g., *WHO, 2016*). Nonetheless, the model herein, validated (largely qualitatively) against field data on a post-honeymoon epidemic in 1988–89, has lessons for measles vaccination policy today. Catch-up SIAs can be an effective way to increase population immunity, especially in areas where difficulties in cold chain maintenance result in lower average vaccine efficacy. However, SIAs should cast a wide net, not only trying to vaccinate infants age 6 months–1 year for the first time, but also moving up in age, even up to 10 year-olds (*Clements, 1994*). It should also be noted that SIAs work with routine vaccination, and are not a replacement (*Gay, 2000*; *Berhane et al., 2009*). Due to cold chain breakdowns, many older children may have been previously vaccinated (i.e.,received a shot) without being immunized: e.g., the mean age of infection of 5 (±4.5) years (Fig. 6). The increasing use of bivalent (measles-rubella) vaccines throughout much of the world (or trivalent measles-mumps-rubella or quadrivalent measles-mumps-rubella-varicella) is all the more reason to cast a wide net in SIAs, since these vaccines also confer protection against rubella (which undergoes similar age shifts), and thus prevent congenital rubella syndrome.

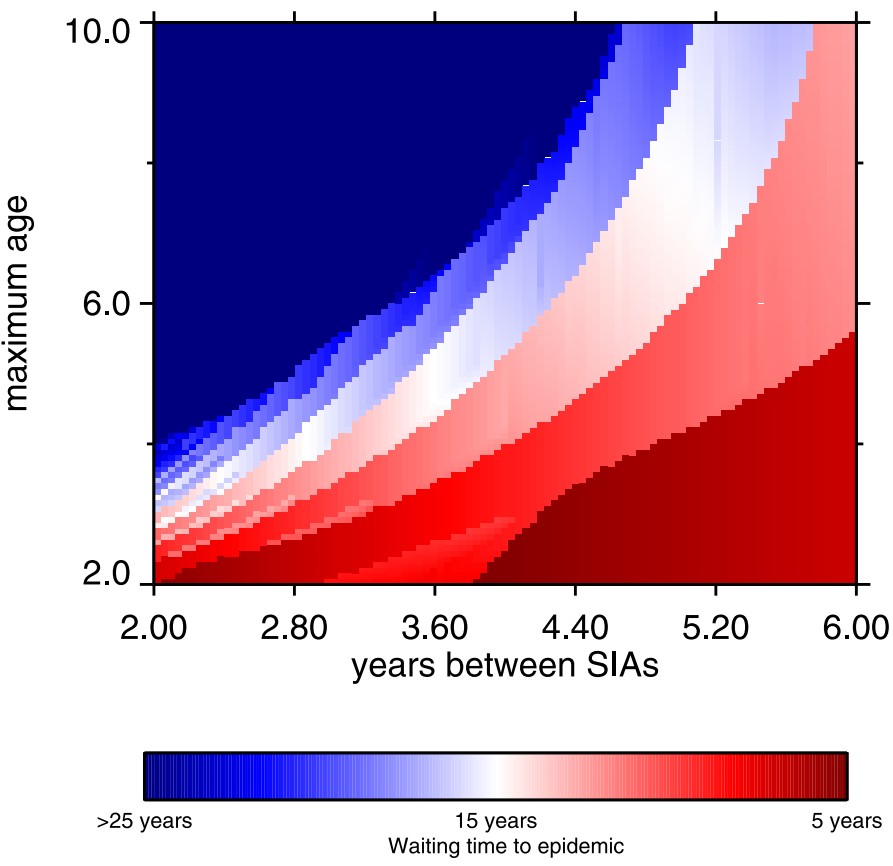

**Figure 9** **Heatmap, waiting time to occurrence of the first post-honeymoon epidemic, frequency of SIA and maximum age of SIA coverage.** The maximum age refers to the target ages of the SIAs; the minimum age is fixed at 9 months. From 10,000 model runs.

This work has several limitations. The assumption that $\beta$ does not vary by age was made due to the lack of empirical data from which to estimate age-dependent transmission parameters. Thus, we use universal mixing as an approximation, which should be feasible for a pathogen as contagious as the measles virus. The qualitative fit of the model to the data is evidence of this. As noted, another limitation is the way the model handles maternal antibody protection. A sudden drop at 6 months is an approximation (*Williams, Cutts & Dye, 1995*). In the observed data, the number of cases below one year is low (5%), so the approximation is an adequate fit in this case. Another limitation is that the model ignores introduction of measles virus from neighboring districts. In real populations, importation of cases alters the epidemiology from the purely endogenously-generated dynamics of the model.

While seasonal forcing has been used in other settings (e.g., *Ferrari et al., 2008*), this model does not include it. It seems to us that the transmission of measles is not significantly seasonally forced in rural Burundi in the presence of vaccination. Burundi has two wet seasons, February to May, and September to November (*Republique du Burundi, 2006*). The observed epidemiology (Fig. 3) does not nest into this seasonality, and the model as it stands fits the data well. This is not to say that adding seasonal terms to Eq. (6) would

degrade the model, but at least to first order, they are not necessary. Elaboration of the model with seasonal forcing would be a potential improvement.

The model was designed for a specific region (Muyinga sector, Burundi), using demography (mortality and birth rate) and size tailored to the region, as well as a force of infection drawing on a literature review of sub-Saharan African serosurveys. The results were interpreted with respect to an empirical data set. In order to preserve population growth as demographically-stable process (in the sense of *Coale, 1972*), we ignored measles mortality, in favor of using life tables that are a fit to available mortality data (ostensibly including measles mortality). This ignores long-run feedback effects of epidemics (*John, 1990b*).

The main policy recommendation is that to avoid measles epidemics, SIAs should be used in addition to routine vaccination, which cannot keep up with the increase in susceptibles caused by population growth. This applies to settings in which vaccine efficacy is less than 95%. Where cold chain challenges or vaccine coverage shortfalls are prevalent, SIAs should be used to bolster immunization. The greater the hurdles to routine vaccination, the more important is the role of supplementary immunization activities. Moreover, there is a age-frequency trade-off for these SIAs, in which less frequent vaccination campaigns must target older children (Fig. 9).

Measles continues to be a challenge in sub-Saharan Africa; although the data we analyze come from an epidemic in Burundi from 25 years ago, the lessons from the present analysis are still applicable today. For instance, measles outbreaks are a current public health problem in the Democratic Republic of Congo, a border nation of Burundi (*Grout et al., 2013*; *Maurice, 2015*; *Scobie et al., 2015*; *Restrepo-Méndez et al., 2016*), as well as elsewhere in sub-Saharan Africa (*Luquero et al., 2011*; *Minetti et al., 2013*). The use of SIAs to achieve measles control is not a novel idea; existing policy recognizes their importance (e.g., *Measles & Rubella Initiative, 2016*). However, as outbreaks continue, mathematical models can help to refine vaccine policy.

## ACKNOWLEDGEMENTS

We thank Robert Chen of the Centers for Disease Control and Prevention for the data. We especially thank Felicity Cutts and Nigel Gay for advice. We thank seminar audiences at Harvard, UC Irvine, Ohio State, and the World Health Organization for feedback, as well as the discussant (Kenneth Wachter) and conference audience at the Population Association of America annual meeting. We also thank the PeerJ referees (Etienne Gignoux, Shaun Truelove, and one anonymous reviewer), for extremely helpful and constructive comments.

### Funding

The authors received no funding for this work.

### Competing Interests

Andrew Noymer is an Academic Editor for PeerJ.

## Author Contributions

- Katelyn C. Corey performed the experiments, analyzed the data, wrote the paper, prepared figures and/or tables, reviewed drafts of the paper.
- Andrew Noymer conceived and designed the experiments, performed the experiments, analyzed the data, wrote the paper, prepared figures and/or tables, reviewed drafts of the paper.

## Data Availability

Raw data and code are supplied as Supplementary Files.

## Supplemental Information

Supplemental information for this article can be found online at http://dx.doi.org/10.7717/peerj.2476#supplemental-information.

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
