# Peer review of "A ‘post-honeymoon’ measles epidemic in Burundi: mathematical model-based analysis and implications for vaccination timing"

_PeerJ, doi:10.7717/peerj.2476_

## Round 0.1 · original submission · Minor Revisions

Two of the reviews contain detailed suggestions that should be helpful to you in revising the manuscript. As you are aware, PeerJ does not place great emphasis on novelty in making publication decisions, but you may wish to clarify the novel contributions of your work to enhance its value to readers as suggested by Reviewer # 3.

·

Basic reporting

Figure 5 : The scale of the incidence is not shown

Experimental design

Line 142 : In general program vaccination target children from 9 to 11 month.

Validity of the findings

Figure 7 and figure 8 show the intervals between epidemics. These are very interesting findings, though the model assumes no exogenous occurrence of infection. An exogenous infection is a likely event. Either this should be noted in the limitation, or exogenous infection should be integrated in the model.

Line 208-221 :
Seroconversion following vaccination is better in children older than 12 month than in children from 9 to 12 months (from 80% to 95%). The better effectiveness of catch up campaigns might be due to the higher mean age of children vaccinated. Assuming that the better effectiveness of catch campaign is due to better coverage and a better cold chain is a debatable assumption. Indeed, during vaccination campaign, the short duration of the campaign can make difficult for families to vaccinate their children. Moreover, the management of a massive cold chain poses logistic challenges that are not always properly solved. Program vaccination needs a smaller cold chain and offer a longer period to families to bring their children for vaccination. In response to the lower seroconversion rate in young child, A 2 doses vaccination is often recommended in program vaccinations. Usually catch up vaccination target children from 6 or from 9 month. Therefore I don’t think it is wise to recommend a catch up campaign that exclude children younger than 24 months.


Line 249:
I agree that the qualitative fit of the model to the data shows that the model can make the assumption that Beta does not vary by age. However, I don’t find relevant to explain it by the “structure of the African rural Societies” (or give references).

Line 263 :
Could you explain why seasonal forcing is not relevant in Burundi

Additional comments

The article is interesting and clear.
I agree with the main recommendation, however I suggest to model a catch up campaign targeting children from 9 month old.

I think it would be interesting in the future to apply this model in settings that faced measles epidemics every 4 years (like Democratic Republic of Congo).

Reviewer 2 ·

Basic reporting

This paper analyzes a post-honeymoon measles outbreak in the 1989-1999 Muyinga sector (Burundi) by using a partial differential equation epidemic model. The results suggest campaigns should be used regularly to supplement program vaccination. The manuscript is well written, and the result seems interesting. I would like to recommend it for publication in PeerJ.

Experimental design

No comments.

Validity of the findings

No comments.

Additional comments

The following are my suggestions.
1. On Page 3: explain the model formulation.
2. Line 82: is there a boundary condition on right when a=omega (where omega is the oldest age)?

·

Basic reporting

Overall, the paper was well written. It is at the same time concise and complete in thought. There are minor edits that should to made to improve understanding, but not critical to the understanding of the writing overall.

The introduction and background are complete, relevant, and build a complete story for the purpose of this study. The references and background information demonstrate in-depth literature research, and sources cited were an appropriate mix of historical and recent publications. Overall it is well referenced.
The structure conforms to that of PeerJ standards, and raw data and code are supplied.

Figures are relevant and of high quality. Figure descriptions need more information regarding to what certain axes are referring. This particularly pertains to figures 5-7. After fully reading the manuscript, I am assuming that ‘time’ refers to time since vaccination introduction, though it’s not clear. Figures should be essentially stand-alone, so these descriptions should be improved.

In the field of vaccine-preventable diseases, the standard phrasing for vaccination types is “routine vaccination” and “supplemental immunization activities” aka “SIA”. I would recommend revising the manuscript to use this terminology.

Experimental design

The experimental design of the manuscript is original and appropriate. While the concept and model are not novel, their application is. Similarly, while the research question is not novel, and the conclusions to this question, that of the need for vaccination campaigns and broad age ranges of these campaigns are widely accepted, the exact interval of these campaigns and specific age ranges are unique findings. I would recommend explicitly stating this as a primary goal of this study.

The methods are described in sufficient detail for both understanding and replication. The model and relevant equations appear to be valid. Evidence for particular parameter or model decisions are well documented and referenced. Sensitivity analysis, though not explicitly stated, were performed with sufficient consideration and scientific rigor.

The conduct of this study appears to be scientifically and ethically sound.

Validity of the findings

The data used are robust and statistically sound. The majority of data are from publicly available sources, and that which are not are provided. However, these provided data (measles and chicken pox incidence) are in a format that is not easily understood, and should be resubmitted in a standard format with columns for year/month and incidence.

A major finding of this study is the impact of vaccination on increasing the average age of infection. While this is already a widely accepted occurrence, this study adds important experimental evidence and quantification of this phenomenon. A second major finding is that of the need for either more frequent campaigns or broader age range to prevent post-honeymoon outbreaks. This again is already a widely held concept, yet this study provides beneficial quantitative demonstration of this occurrence and estimates of the age range/campaign interval trade off.

As these are neither novel overall findings nor recommendations, revision is warranted to better demonstrate how these particular findings are important. The policy recommendations from the WHO and other sources are already that vaccination campaigns are necessary in countries with sub-optimal vaccination coverage. Furthermore, given that this study pertains to data from the 1980’s these findings and not directly of value to Burundi. However, the great value of this work is the demonstration, quantitatively, of these phenomena and the estimation of the necessary age ranges and timing intervals campaigns. These should be focused on more.

Additional comments

Overall, this is a well conducted study that provides important evidence to support and quantify occurrences that challenge the current efforts to control and eliminate measles and other vaccine-preventable diseases. This manuscript is generally well written, the science is sound, and it merits publication.

I would like to see some edits prior to publication. These are as follows:

Major Edits:
1. Revise the first and second paragraph of the Discussion. The first paragraph needs to immediately restate the purpose or question of the study, and then what this study’s findings contribute. As it is written, I found myself searching for conclusions relevant to the findings, only to be inundated with more citations and irrelevant information (lines 227-231, sentence starting with “One of the insights…” through end of the paragraph are not immediately relevant the findings of this study so should be either moved or deleted).
2. Increase the focus of the discussion and other parts of the paper on the estimation of necessary campaign age ranges and timing intervals.
3. Change terminology to use “routine vaccination” and “supplemental immunization activity” or “SIA”.

Minor Edits:
- Line 11-12: Restate in the active voice.
- Line 12: Delete comma after ‘vaccination’
- Line 34-37 (“An outbreak of …”): Does not seem relevant or needs to be rephrased.
- Line 46 - “such as ours”: Change to “like ours”. This model has not influenced policy yet.
- Line 72 - “Moreover, the life table…”: While the life table does include deaths, which includes measles deaths, I assume (based on your model above) that you attribute the same death rate to all compartments of the model, so you are actually attributing a slightly reduced death rate to those in the c or z compartments, and slightly increased rate to the other compartments. Probably best here to just delete this sentence.

---

## Round 0.2 · accepted · Accept

Three reviewers have looked at the revised version and indicated that they are satisfied and have no further critiques.

·

Basic reporting

no comments

Experimental design

no comments

Validity of the findings

no comments

Additional comments

Great article, thank you.

Reviewer 2 ·

Basic reporting

The revised manuscript is very clearly written and the results are interesting. The referee strongly recommend it for publication.

Experimental design

The referee doesn't have further comments.

Validity of the findings

No comments.

·

Basic reporting

No Comments

Experimental design

No Comments

Validity of the findings

No Comments

Additional comments

The manuscript has been revised appropriately and should be accepted for publication.

---

## Author Rebuttal · Round 0.2

# Response to Referees

## PeerJ Manuscript #11486

## 9 August 2016

The referee comments are *in italics* and our responses are interleaved in roman text.

All page and line numbers in our responses refer to the revised-and-resubmitted version.

## Editor (Arthur S. Sherman)

*Two of the reviews contain detailed suggestions that should be helpful to you in revising the manuscript. As you are aware, PeerJ does not place great emphasis on novelty in making publication decisions, but you may wish to clarify the novel contributions of your work to enhance its value to readers as suggested by Reviewer # 3.*

We thank the editor for handling our manuscript. The reviewers' comments are constructive, and we feel that our revised manuscript is greatly improved. Our changes are documented herein. We have acknowledged the referees' input in the acknowledgments.

## Reviewer 1 (Etienne Gignoux)

Basic reporting

*Figure 5 : The scale of the incidence is not shown*

This has been fixed. The scale is now shown, and appropriate text has been added to the figure caption, providing further explanation. The referee will note that the vertical axis is now labeled "prevalence" — it was an error to have labeled it "incidence". The figure caption in the original submission was correct: this is a prevalence surface. We thank the referee for this suggestion, which has been implemented and moreover which prompted us to notice the labeling error, which of course we regret and are glad to have corrected. (Figure 5, page 8.)

### Experimental design

*Line 142 : In general program vaccination target children from 9 to 11 month.*

The paper has been changed accordingly (line 147). We thank the reviewer for this improvement.

### Validity of the findings

*Figure 7 and figure 8 show the intervals between epidemics. These are very interesting findings, though the model assumes no exogenous occurrence of infection. An exogenous infection is a likely event. Either this should be noted in the limitation, or exogenous infection should be integrated in the model.*

This is a good point, and we have added it to the limitations (lines 249–50).

*Line 208–221: Seroconversion following vaccination is better in children older than 12 month than in children from 9 to 12 months (from 80% to 95%). The better effectiveness of catch up campaigns might be due to the higher mean age of children vaccinated. Assuming that the better effectiveness of catch campaign is due to better coverage and a better cold chain is a debatable assumption. Indeed, during vaccination campaign, the short duration of the campaign can make difficult for families to vaccinate their children. Moreover, the management of a massive cold chain poses logistic challenges that are not always properly solved. Program vaccination needs a smaller cold chain and offer a longer period to families to bring their children for vaccination. In response to the lower seroconversion rate in young child, A 2 doses vaccination is often recommended in program vaccinations. Usually catch up vaccination target children from 6 or from 9 month. Therefore I don't think it is wise to recommend a catch up campaign that exclude children younger than 24 months.*

We thank the referee for this thoughtful comment. We agree wholeheartedly that this is a better explanation for the improved effectiveness of catch up campaigns, and the paper has been changed accordingly (line 219).

*Line 249: I agree that the qualitative fit of the model to the data shows that the model can make the assumption that Beta does not vary by age. However, I don't find relevant to explain it by the "structure of the African rural Societies" (or give references).*

We have removed speculation as to the idiosyncratic nature of African rural societies, and stuck to the basics, namely that we are making an approximation. We thank the referee for this helpful comment.

*Line 263 : Could you explain why seasonal forcing is not relevant in Burundi*

We have added an explanation for this (¶, lines 251–57), and moreover we have noted omission of seasonal forcing as a limitation. We think that our treatment of this in the revised paper is an improvement, and we thank the referee for prompting it.

Comments for the Author

*The article is interesting and clear. I agree with the main recommendation, however I suggest to model a catch up campaign targeting children from 9 month old.*

We thank the reviewer for his overall enthusiasm as well as for the specific comments.

In response to the referee's useful suggestion, we have re-run the model, changing the campaign vaccination (viz., SIA) to target 9 months–X years of age, not 2–X years where X refers to a variable upper bound (cf. figure 9). Naturally, this changes the results somewhat, and the text has been changed accordingly. See lines 151 and 225.

*I think it would be interesting in the future to apply this model in settings that faced measles epidemics every 4 years (like Democratic Republic of Congo).*

This is an excellent point and we have added it to the conclusion, lines 274–77.

The reviewer may also note that some of our changes in response to the third referee are along the same lines.

# Reviewer 2 (Anonymous)

## Basic reporting

*This paper analyzes a post-honeymoon measles outbreak in the 1989–1999 Muyinga sector (Burundi) by using a partial differential equation epidemic model. The results suggest campaigns should be used regularly to supplement program vaccination. The manuscript is well written, and the result seems interesting. I would like to recommend it for publication in PeerJ.*

We thank the referee for the positive reaction.

## Experimental design

*No comments.*

## Validity of the findings

*No comments.*

## Comments for the Author

*The following are my suggestions.*

*1. On Page 3: explain the model formulation. 2. Line 82: is there a boundary condition on right when a=omega (where omega is the oldest age)?*

We have enhanced the explanation of the model (cf. especially lines 69–71)

Yes, there is boundary condition on the right; everyone dies at age $\omega$. This has now been noted, i.e., "$\mu(\omega) = \infty$", along with the other boundary condition. We hope this is the correct notation; technically, one could argue whether $\mu(\cdot)$ can equal infinity or if it "approaches" infinity. In any case, we think the new text will be understood. And, yes, clearly, there is a boundary condition on the right. We thank the referee for this. Line 86.

## Reviewer 3 (Shaun Truelove)

### Basic reporting

*Overall, the paper was well written. It is at the same time concise and complete in thought. There are minor edits that should to made to improve understanding, but not critical to the understanding of the writing overall.*

We thank the reviewer for this remark as well as his close read of the paper.

*The introduction and background are complete, relevant, and build a complete story for the purpose of this study. The references and background information demonstrate in-depth literature research, and sources cited were an appropriate mix of historical and recent publications. Overall it is well referenced. The structure conforms to that of PeerJ standards, and raw data and code are supplied.*

We thank the reviewer; we tried, within reasonable limits, to give both an historical perspective on measles models and an up to date lit review, and we are grateful that our efforts were well-received.

*Figures are relevant and of high quality. Figure descriptions need more information regarding to what certain axes are referring. This particularly pertains to figures 5-7. After fully reading the manuscript, I am assuming that 'time' refers to time since vaccination introduction, though it's not clear. Figures should be essentially stand-alone, so these descriptions should be improved.*

We have changed the figure captions. We agree they are clearer now, and, in particular, a would-be reader looking at the figures without reading the body text will, we hope, understand a lot better.

Please see also our response to reviewer 1, as regards his comment on figure 5.

*In the field of vaccine-preventable diseases, the standard phrasing for vaccination types is "routine vaccination" and "supplemental immunization activities" aka "SIA". I would recommend revising the manuscript to use this terminology. Experimental design The experimental design of the manuscript is original and appropriate. While the concept and model are not novel, their application is. Similarly, while the research question is not novel, and the conclusions to this question, that of the need for vaccination campaigns and broad age ranges of these campaigns*

*are widely accepted, the exact interval of these campaigns and specific age ranges are unique findings. I would recommend explicitly stating this as a primary goal of this study.*

We thank the referee for this assessment and the suggestion. We have changed the terminology throughout. We have added an emphasis on the SIA frequency and age range (lines 212–227), and a new figure 9 dealing with such.

*The methods are described in sufficient detail for both understanding and replication. The model and relevant equations appear to be valid. Evidence for particular parameter or model decisions are well documented and referenced. Sensitivity analysis, though not explicitly stated, were performed with sufficient consideration and scientific rigor.*

Thank you.

*The conduct of this study appears to be scientifically and ethically sound.*

Thank you.

## Validity of the findings

*The data used are robust and statistically sound. The majority of data are from publicly available sources, and that which are not are provided. However, these provided data (measles and chicken pox incidence) are in a format that is not easily understood, and should be resubmitted in a standard format with columns for year/month and incidence.*

We have re-uploaded the data. We have added column labels, and a codebook. We have also converted the files from Unix-style ASCII to Win/Mac-style ASCII, which we suspect was part of the problem.

*A major finding of this study is the impact of vaccination on increasing the average age of infection. While this is already a widely accepted occurrence, this study adds important experimental evidence and quantification of this phenomenon. A second major finding is that of the need for either more frequent campaigns or broader age range to prevent post-honeymoon outbreaks. This again is already a widely held concept, yet this study provides beneficial quantitative demonstration of this occurrence and estimates of the age range/campaign interval trade off.*

We thank the referee for the close read of our paper. Thanks to his advice we have produced a manuscript that better stresses the novel aspects.

*As these are neither novel overall findings nor recommendations, revision is warranted to better demonstrate how these particular findings are important. The policy recommendations from the WHO and other sources are already that vaccination campaigns are necessary in countries with sub-optimal vaccination coverage. Furthermore, given that this study pertains to data from the 1980's these findings and not directly of value to Burundi. However, the great value of this work is the demonstration, quantitatively, of these phenomena and the estimation of the necessary age ranges and timing intervals campaigns. These should be focused on more.*

We have changed the paper according to the referee's comment; cf. especially figure 9 and the associated discussion. We thank the referee for how much these comments have prompted us to change the paper for the better.

## Comments for the Author

*Overall, this is a well conducted study that provides important evidence to support and quantify occurrences that challenge the current efforts to control and eliminate measles and other vaccine-preventable diseases. This manuscript is generally well written, the science is sound, and it merits publication.*

We thank the referee, and we really feel that his comments have improved the paper.

I would like to see some edits prior to publication. These are as follows:

## Major Edits:

*1. Revise the first and second paragraph of the Discussion. The first paragraph needs to immediately restate the purpose or question of the study, and then what this study's findings contribute. As it is written, I found myself searching for conclusions relevant to the findings, only to be inundated with more citations and irrelevant information (lines 227–231, sentence starting with "One of the insights..." through end of the paragraph are not immediately relevant the findings of this study so should be either moved or deleted).*

The former first paragraph of the discussion has been removed. Most of it was redundant, and these parts have been deleted. A few bits have been moved to more appropriate places. The referee is correct that this (former) paragraph was in the wrong place and we thank him for the good advice.

*2. Increase the focus of the discussion and other parts of the paper on the estimation of necessary campaign age ranges and timing intervals.*

We have done this. Again, we thank the referee for giving the paper a firm push in the right direction.

*3. Change terminology to use "routine vaccination" and "supplemental immunization activity" or "SIA".*

Done.

## Minor Edits:

*- Line 11-12: Restate in the active voice.*

Done.

*- Line 12: Delete comma after 'vaccination'*

Done.

*- Line 34-37 ("An outbreak of..."): Does not seem relevant or needs to be rephrased.*

The reviewer is correct that the sentence was not essential to the paper. It has been deleted.

*- Line 46 - "such as ours": Change to "like ours". This model has not influenced policy yet.*

Good point. Done. Line 48.

*- Line 72 - "Moreover, the life table...": While the life table does include deaths, which includes measles deaths, I assume (based on your model above) that you attribute the same death rate to all compartments of the model, so you are actually attributing a slightly reduced death rate to those in the c or z compartments, and slightly increased rate to the other compartments. Probably best here to just delete this sentence.*

The sentence is deleted.